# A feedback loop between nonsense-mediated decay and the retrogene *DUX4* in facioscapulohumeral muscular dystrophy

Qing Feng[1,2], Lauren Snider[3], Sujatha Jagannathan[1,2,3], Rabi Tawil[4], Silvère M van der Maarel[5], Stephen J Tapscott[3,6]*, Robert K Bradley[1,2]*

[1]Computational Biology Program, Public Health Sciences Division, Fred Hutchinson Cancer Research Center, Seattle, United States; [2]Basic Sciences Division, Fred Hutchinson Cancer Research Center, Seattle, United States; [3]Human Biology Division, Fred Hutchinson Cancer Research Center, Seattle, United States; [4]Department of Neurology, University of Rochester, Rochester, United States; [5]Department of Human Genetics, Leiden University Medical Center, Leiden, Netherlands; [6]Department of Neurology, University of Washington, Seattle, United States

**Abstract** Facioscapulohumeral muscular dystrophy (FSHD) is a muscular dystrophy caused by inefficient epigenetic repression of the D4Z4 macrosatellite array and somatic expression of the *DUX4* retrogene. DUX4 is a double homeobox transcription factor that is normally expressed in the testis and causes apoptosis and FSHD when misexpressed in skeletal muscle. The mechanism(s) of DUX4 toxicity in muscle is incompletely understood. We report that DUX4-triggered proteolytic degradation of UPF1, a central component of the nonsense-mediated decay (NMD) machinery, is associated with profound NMD inhibition, resulting in global accumulation of RNAs normally degraded as NMD substrates. DUX4 mRNA is itself degraded by NMD, such that inhibition of NMD by DUX4 protein stabilizes DUX4 mRNA through a double-negative feedback loop in FSHD muscle cells. This feedback loop illustrates an unexpected mode of autoregulatory behavior of a transcription factor, is consistent with 'bursts' of *DUX4* expression in FSHD muscle, and has implications for FSHD pathogenesis.

*For correspondence: stapscot@ fhcrc.org (SJT); rbradley@fhcrc.org (RKB)

**Competing interests:** The authors declare that no competing interests exist.

**Reviewing editor**: Rachel Green, HHMI, Johns Hopkins University School of Medicine, United States

## Main text

Facioscapulohumeral muscular dystrophy (FSHD) is typically an adult-onset muscular dystrophy characterized by muscle weakness initially affecting the face (facio), shoulders (scapulo), and upper arms (humeral). FSHD is caused by decreased epigenetic repression of the D4Z4 macrosatellite array in the subtelomeric region of chromosome 4q, due to either D4Z4 repeat contractions (*Lemmers et al., 2010*) or mutations affecting *trans*-acting epigenetic regulators of the D4Z4 repeat such as SMCHD1 (*Lemmers et al., 2012*), which results in the misexpression of DUX4 mRNA in skeletal muscle and possibly other somatic tissues. *DUX4* encodes a double homeobox transcription factor that activates germline genes and repetitive elements (*Geng et al., 2012*) and causes apoptosis and atrophic myotube formation when misexpressed in skeletal muscle (*Kowaljow et al., 2007*; *Vanderplanck et al., 2011*; *Wallace et al., 2011*; *Mitsuhashi et al., 2012*). *DUX4* is expressed in only a small fraction of nuclei (*Snider et al., 2010*), likely due to occasional 'bursts' of *DUX4* expression. However, the mechanism(s) regulating *DUX4* expression and toxicity remain incompletely understood.

We previously ectopically expressed *DUX4* in immortalized (54-1) and primary (MB135) myoblasts and used RNA-seq to identify coding genes, repetitive elements, and non-coding RNAs induced by DUX4 (*Young et al., 2013*). Further analysis of this data showed that *DUX4* expression also resulted in

**eLife digest** Genes are sequences of DNA that contain instructions for the cell that must be carefully controlled because it is not always appropriate or safe for these instructions to be followed. When genes are active, copies of the DNA are made using molecules of ribonucleic acid (RNA) and these can then be used as templates to make proteins.

One way genes can be controlled is by adding small chemical tags that mark them out to be activated or deactivated, known as epigenetic control. The muscle disease facioscapulohumeral muscular dystrophy (FSHD) is caused by the loss of the chemical tags that normally keep certain genes switched off in many cell types. One of these genes is *DUX4*, which in healthy males is normally only active in the testes, but in FSHD patients it is also active in other parts of the body.

Another way to control genes is by nonsense-mediated decay, where incorrect or incomplete RNA molecules are destroyed before they can be used to make defective proteins. In this study, Feng et al. show that when *DUX4* is activated following the failure of epigenetic control in FSHD patients, the effectiveness of nonsense-mediated decay is also reduced. This results in the build-up of incorrect RNA molecules inside muscle cells, which can harm the cell. In fact, 13% of the incorrect RNA molecules that are normally destroyed in cells were found at higher levels when *DUX4* was active.

To investigate how *DUX4* could work, Feng et al. focused on another gene called *UPF1* because cells without the protein encoded by this gene have similar defects in nonsense-mediated decay as cells with active *DUX4*. No difference was found in how often the *UPF1* gene is activated in FSHD cells and normal cells. However, the amount of the protein encoded by *UPF1* was lower in cells with FSHD than in normal muscle cells. The experiments show that the protein encoded by *UPF1* is broken down as a result of the activation of the *DUX4* gene, leading to problems with nonsense-mediated decay, which may result in the worsening of FSHD symptoms.

The twist in the tale is that *DUX4* itself is also controlled by nonsense-mediated decay under normal circumstances. Therefore, in diseased cells, a failure in epigenetic control allows *DUX4* to prevent its own destruction by tampering with nonsense-mediated decay. These findings offer new insights into the role of the *DUX4* gene in FSHD. The next step is to test whether these defects in nonsense-mediated decay can explain any of the symptoms of FSHD, such as muscle inflammation.

the increased abundance of many coding RNA isoforms containing premature translation termination codons upstream of splice junctions. These isoforms, which are predicted substrates for degradation by nonsense-mediated decay (NMD), were present at very low levels in control myoblasts. Following *DUX4* expression, however, many such predicted NMD substrates increased in abundance and in many cases became the predominant mRNA product of the parent gene. For example, an isoform of the *SRSF3* gene containing a well-characterized NMD-inducing cassette exon (*Lareau et al., 2007*; *Ni et al., 2007*) was present at low levels prior to *DUX4* expression but became the dominant isoform thereafter in both 54-1 and MB135 cells (*Figure 1A–B*).

To determine whether increased levels of such normally degraded mRNAs were associated with reduced NMD efficiency, we used an exogenous reporter system. We transfected plasmids encoding either the wild-type β-globin open reading frame or β-globin with a premature termination codon that induces degradation by NMD (*Zhang et al., 1998*). Relative levels of the β-globin NMD substrate were twofold higher in *DUX4*-expressing vs control myoblasts, indicating that NMD is indeed compromised by DUX4 (*Figure 1C*).

We then determined how reduced NMD efficiency affected global levels of predicted NMD substrates. Restricting to cassette exon splicing events where one isoform, but not both, was a predicted NMD substrate, we found that ~13% of such predicted NMD substrates increased following *DUX4* expression, while ~1.6% decreased, in 54-1 cells (*Figure 1D*). Impaired NMD also caused accumulation of aberrant mRNAs resulting from mis-splicing or incomplete splicing, which are common byproducts of the stochastic nature of the splicing process (*Weischenfeldt et al., 2012*). We identified and quantified alternative splicing of annotated constitutive junctions, finding that ~13% of such junctions exhibited increased aberrant splicing in *DUX4*-expressing vs control cells, while only ~0.25% exhibited decreased aberrant splicing (*Figure 1E*). The vast majority of these novel products of annotated constitutive junctions were present at very low or undetectable levels in control 54-1 myoblasts.

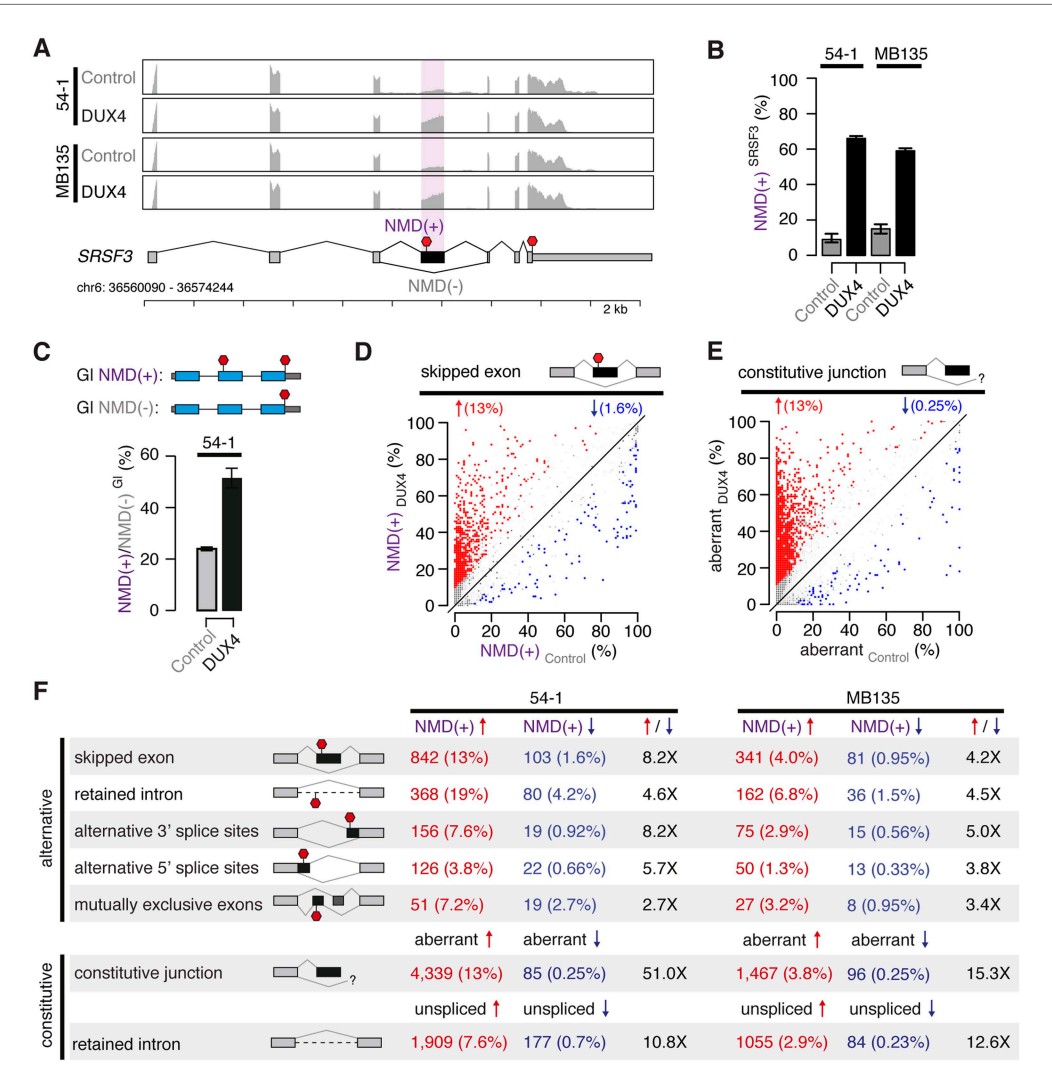

**Figure 1**. *DUX4* expression inhibits nonsense-mediated decay. (**A**) RNA-seq read coverage of the *SRSF3* gene in control and DUX4-expressing myoblasts. 54-1, immortalized myoblasts; MB135, primary myoblasts. Purple shading, NMD-inducing cassette exon. Red stop sign, termination codon. (**B**) Inclusion of the premature termination codon-containing cassette exon of *SRSF3* illustrated in (**A**). Error bars, 95% confidence intervals as estimated by MISO (*Katz et al., 2010*). (**C**) Relative levels of transcripts produced from NMD reporter plasmids encoding either premature termination codon-containing (top) or normal (bottom) β-globin (Gl). Bar plot illustrates the ratio NMD(+)/NMD(−) of transcripts from the NMD(+) and NMD(−) constructs. (**D**) Isoform ratios of predicted NMD substrates generated by cassette exon alternative splicing in control and *DUX4*-expressing myoblasts (54-1 cells). Red/blue, cassette exons exhibiting increases/decreases of ≥10% in isoform ratios for the isoforms that are predicted NMD substrates. (**E**) Isoform ratios of mis-spliced isoforms of annotated constitutive splice junctions generated by abnormal 5' and 3' splice site recognition (54-1 cells). Color as in (**D**). (**F**) Global increases and decreases in relative levels of predicted NMD substrates generated by differential splicing. Annotated alternative splicing events are illustrated in upper panel, and alternative splicing and intron retention of annotated constitutively spliced junctions are illustrated in lower panel. Up/down arrows, percentages of predicted NMD substrates generated by alternative splicing exhibiting increases/decreases of ≥10% in isoform ratios in *DUX4*-expressing vs control cells. Enrichment for increased vs decreased levels of NMD substrates indicated in columns three and six.

The following figure supplement is available for figure 1:

**Figure supplement 1**. DUX4-induced NMD inhibition is not a side effect of DUX4 toxicity.

We next extended this analysis to all classes of splicing events, including mis-splicing and intron retention of constitutive splice junctions. *DUX4* expression caused increased levels of predicted NMD substrates for all classes of splicing events in both 54-1 and MB135 cells (*Figure 1F*). These increases were generally more extreme in 54-1 than in MB135 cells, likely due to the ~15-fold higher *DUX4* expression achieved in 54-1 vs MB135 cells as well as the longer time period allowed for infection (48 hr vs 24 hr).

High levels of NMD substrates in *DUX4*-expressing cells were not simply a side effect of DUX4-induced apoptosis. *TP53* knock-down (KD) prevented apoptosis following *DUX4* expression in normal myoblasts, confirming previous reports that DUX4 toxicity is p53-dependent (*Wallace et al., 2011*). However, *TP53* KD did not prevent DUX4-induced NMD inhibition (*Figure 1—figure supplement 1*).

DUX4 could potentially inhibit NMD by transcriptionally repressing components of the NMD machinery. However, no UPF or SMG NMD factors exhibited decreased mRNA levels following *DUX4* expression, and most were up-regulated by two- to fourfold (*Figure 2A*). This expression pattern was reminiscent of a recent report that mRNA levels of most NMD factors increase following the knock-down of *UPF1*, encoding a central component of the NMD machinery (*Huang et al., 2011*). Therefore, we hypothesized that UPF1 mRNA and protein levels might be decoupled in *DUX4*-expressing cells. We measured levels of UPF1, which was not transcriptionally up-regulated in *DUX4*-expressing cells, and UPF3B and SMG7, which were transcriptionally up-regulated in response to DUX4. UPF1 protein levels were markedly lower in *DUX4*-expressing myoblasts than in control myoblasts, as were SMG7 levels, although to a lesser extent. In contrast, UPF3B levels were unaffected by *DUX4* expression (*Figure 2B*).

To determine whether decreased UPF1 temporally correlates with DUX4-induced inefficient NMD, we conducted a time course following *DUX4* expression in myoblasts. DUX4 was robustly detectable 12–14 hr after lentiviral infection, coincident with the beginning of a sharp decrease in UPF1 levels (*Figure 2C–D*). SMG7 showed a more modest decrease through the time course, while UPF3B levels were relatively constant. NMD substrates produced from the β-globin reporter, as well as endogenously produced from the *SRSF2* and *SRSF3* genes, exhibited increased levels 12–14 hr after lentiviral expression (*Figure 2E–F*). The close temporal coupling between DUX4 protein production, decreased UPF1 levels, and increased levels of both endogenous and exogenous NMD substrates suggests that insufficient levels of UPF1—and perhaps additional NMD machinery components such as SMG7—may contribute to inefficient NMD in *DUX4*-expressing cells.

The rapid decrease in UPF1 levels that we observed suggested that DUX4 might trigger UPF1 degradation. To test this, we treated *DUX4*-expressing or control myoblasts with MG132 to inhibit the proteasome. MG132 treatment restored normal UPF1 levels in *DUX4*-expressing myoblasts, while UPF1 levels in control myoblasts were unaffected (*Figure 2G–H*). As proteasome inhibition inhibits normal translation (*Cowan and Morley, 2004*; *Mazroui et al., 2007*) and therefore NMD, we were unable to test whether the restoration of normal UPF1 levels by proteasomal inhibition rescued NMD. However, the close temporal relationship between the onset of decreased UPF1 levels and increased NMD substrates strongly suggests that UPF1 degradation contributes to NMD inhibition in *DUX4*-expressing cells.

Both *DUX4* isoforms encoding the full-length protein contain a constitutively spliced intron within their 3′ UTRs, rendering them likely NMD substrates (*Figure 3A*). To test this, we used cells isolated from FSHD1 (54-2, which are isogenic to normal 54-1 cells but carry a contracted D4Z4 array) and FSHD2 (MB200) skeletal muscle (*Krom et al., 2012*; *Schoenberg and Maquat, 2012*; *Young et al., 2013*). We knocked down *UFP1* in 54-2 and MB200 myoblasts to 24.3% and 32.4% of normal protein levels, respectively, and differentiated these myoblasts to myotubes to stimulate *DUX4* transcription. DUX4 mRNA was expressed at approximately fourfold higher levels in *UPF1* KD vs control KD myotubes, as was ZSCAN4 mRNA, which is transcriptionally activated by DUX4 (*Figure 3B–D*).

We next sought to determine whether the intron-containing 3′ UTR of *DUX4* contributed to the degradation of DUX4 mRNA by NMD. We created chimeric constructs containing the β-globin open reading frame followed by either the complete *DUX4* 3′ UTR or the *DUX4* 3′ UTR with the second intron removed (*Figure 3E*). We focused on the constitutively spliced second intron within the 3′ UTR because it lies 100 nt downstream of the termination codon, and therefore it is predicted to trigger NMD. Transcripts from the chimeric construct containing the complete *DUX4* 3′ UTR increased twofold following *UPF1* KD in normal myoblasts—a substantial but smaller increase than we observed for the endogenous DUX4 mRNA, perhaps due to the chimeric nature of the β-globin + *DUX4* 3′ UTR construct—while

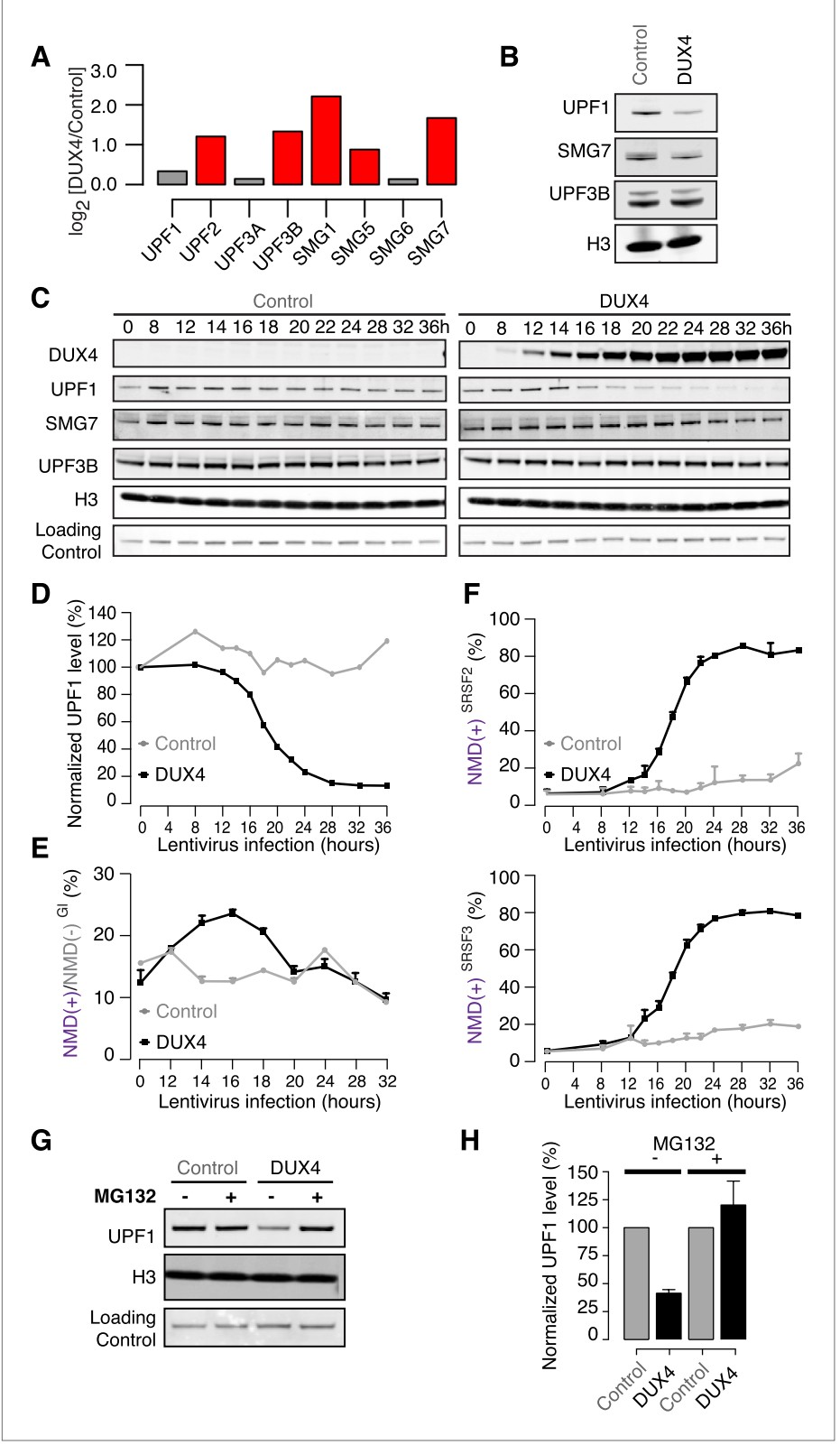

**Figure 2**. DUX4 destabilizes UPF1 via the proteasome. (**A**) Relative mRNA levels of NMD factors in *DUX4*-expressing vs control myoblasts (54-1 cells). Red, up-regulation by ≥1.5-fold. (**B**) Immunoblot NMD factors UPF1, SMG7, and UPF3B in *DUX4*-expressing and control myoblasts (54-1 cells) at 36 hr post-infection. H3, histone H3 (loading control). *Figure 2. Continued on next page*

*Figure 2. Continued*

(**C**) Immunoblot of total protein from a 36-hr time course of *DUX4*-expressing and control myoblasts (54-1 cells). H3, histone H3. Loading Control, a nonspecific band that serves as an additional loading control. (**D**) Quantification of UPF1 protein level from the immunoblot presented in (**C**), normalized to the nonspecific band that serves as a loading control. (**E**) Relative levels of transcripts produced from the NMD(+) and NMD(−) β-globin reporter plasmids. (**F**) Isoform ratios of endogenously produced NMD-degraded isoforms of *SRSF2* and *SRSF3*. Time course identical to (**C**). Error bars, standard deviation. (**G**) Immunoblot of total protein from *DUX4*-expressing and control myoblasts (54-1 cells) treated with the proteasome inhibitor MG132 (10 µM; 8 hr treatment initiated 16 hr after infection with lentiviral expression constructs). Loading control H3, histone 3, has a long half-life (*Toyama et al., 2013*). (**H**) Quantification of UPF1 protein levels from three independent replicates of the immunoblot presented in (**G**), normalized to the nonspecific band that serves as a loading control. Error bars, standard deviation.

transcripts from the construct lacking the second intron of the *DUX4* 3′ UTR increased only 1.5-fold. We conclude that the second intron of the *DUX4* 3′ UTR is important for NMD-induced degradation of the DUX4 mRNA (*Figure 3F*).

*DUX4* exhibits variegated expression in FSHD muscle cells, with only a few percent of nuclei detectable as DUX4+ (*Snider et al., 2010*). Therefore, augmented DUX4 expression following *UPF1* KD in myotubes could be due to increases in DUX4 mRNA in nuclei that are already DUX4+ and/or increases in the fraction of DUX4+ nuclei. Immunostaining of FSHD myotubes revealed that the fraction of DUX4+ nuclei increased from 0.3% to 2.1% following *UPF1* KD, a substantial order-of-magnitude increase (*Figure 3G*). Together, our data show that NMD is an endogenous suppressor of DUX4 mRNA levels that contributes to the very low and variegated expression of *DUX4*, a characteristic feature of FSHD muscle cells.

As *DUX4* expression inhibits NMD and NMD degrades DUX4 mRNA, we hypothesized that *DUX4* and the NMD pathway might participate in a double-negative feedback loop (*Figure 4A*). This feedback loop predicts that DUX4 will indirectly stabilize its own mRNA by inhibiting NMD. To test this, we ectopically expressed DUX4 in FSHD1 and FSHD2 myotubes and measured levels of endogenously transcribed DUX4 mRNA. Ectopic DUX4 expression led to an approximately fivefold increase in endogenously transcribed DUX4 mRNA levels (*Figure 4B*). We next tested whether *DUX4*'s spliced 3′ UTR, which is important for NMD-mediated degradation of DUX4 mRNA, contributed to this increase. We transfected our chimeric β-globin + *DUX4* 3′ UTR reporters into normal myoblasts and ectopically expressed *DUX4*. Levels of the NMD-susceptible construct containing the complete *DUX4* 3′ UTR increased 1.43-fold following ectopic DUX4 expression, while levels of the construct without the second intron of the *DUX4* 3′ UTR exhibited a more modest increase of 1.08-fold (*Figure 4C*). As with the *UPF1* KD experiments, the chimeric construct exhibited more modest effect sizes in these feedback loop experiments than we observed for the endogenous DUX4 mRNA itself.

Together, our data demonstrate that the *DUX4* 3′ UTR targets DUX4 mRNA for NMD and that DUX4-mediated inhibition of NMD results in increased perdurance of the DUX4 mRNA as a possible mechanism of positive autoregulation (*Figure 4D*). It is unclear whether NMD-mediated autoregulation is intrinsic to normal DUX4 function or instead an abnormal consequence of inappropriate DUX4 expression in skeletal muscle. However, it is interesting to consider that this mechanism might contribute to the spreading of *DUX4* expression between adjacent nuclei in a muscle fiber. Because muscle fibers contain arrays of closely spaced nuclei, the expression of DUX4 mRNA from one nucleus will distribute protein to the surrounding nuclei and induce a region of NMD inhibition. If one of the surrounding nuclei subsequently expresses *DUX4*, then that mRNA would be unusually stable due to locally inefficient NMD, thereby facilitating the spread of DUX4 mRNA and protein throughout the fiber.

The close temporal coupling between the onset of DUX4 expression, decreases in UPF1 protein, and increases in NMD substrates (*Figure 2*) strongly suggests that DUX4-mediated degradation of UPF1 contributes to DUX4-induced NMD inhibition. In the absence of a direct mechanistic link between UPF1 degradation and NMD inhibition, we were unable to determine whether insufficient UPF1 protein levels are primarily responsible for DUX4-induced NMD inhibition or instead merely one of the several contributing factors. Nonetheless, as we are unaware of other reports of physiological stimuli triggering rapid UPF1 protein degradation, our data suggest that UPF1 proteolysis constitutes a potential new regulator of cellular NMD efficiency. DUX4 may prove a useful system to gain insight into the biological relevance of this mechanism for altering NMD efficiency. We previously observed

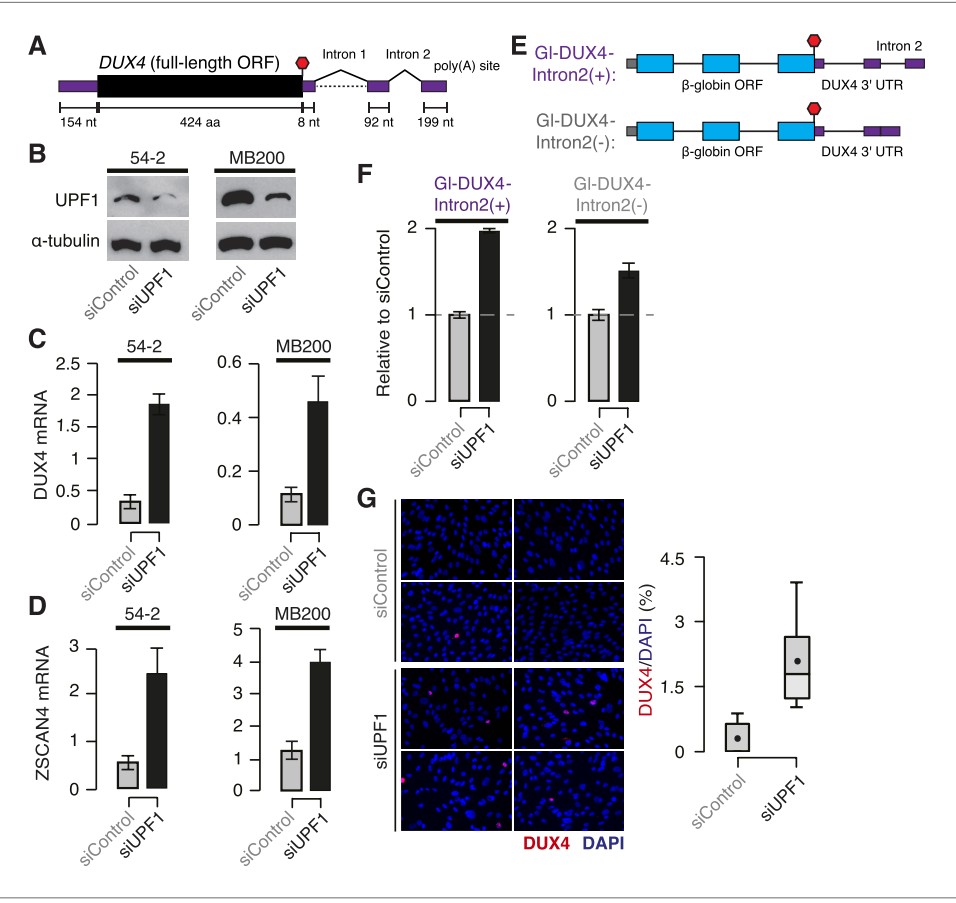

**Figure 3**. DUX4 mRNA is an endogenous NMD substrate. (**A**) Schematic of the DUX4 mRNA. Intron 2, constitutively spliced intron within the 3' UTR. Black, coding sequence; purple, 5' and 3' untranslated regions (UTRs). Red stop sign, termination codon. (**B**) Immunoblot of total protein from FSHD1 (54-2) and FSHD2 (MB200) myoblasts following transfection with a siRNA against *UPF1* or a control non-targeting siRNA. α-tubulin, loading control. (**C**) Levels of DUX4 mRNA following control or *UPF1* knock-down, measured 2 days after the initiation of myogenesis. Error bars, standard deviation. (**D**) Levels of ZSCAN4 mRNA following control or *UPF1* knock-down, measured 2 days after the initiation of myogenesis. Error bars, standard deviation. (**E**) Schematic of chimeric constructs encoding the β-globin opening reading frame (cyan) followed by the *DUX4* 3' UTR (purple) containing (top) or lacking (bottom) the second intron of *DUX4*'s 3' UTR (Intron 2). (**F**) Relative levels of transcripts from the Gl-DUX4-Intron2(+) and Gl-DUX4-Intron2(−) constructs following control or *UFP1* KD in normal myoblasts (54-1 cells). For each construct, data are normalized such that the siControl treatment is 1. Error bars, standard deviation. (**G**) Immunofluorescence with an antibody against DUX4 following control or *UPF1* knock-down, measured 2 days after the initiation of myogenesis in FSHD1 cells (54-2), which was prior to significant fusion. Box plot, percentage of DUX4+ nuclei as estimated by ImageJ (Fiji); n = 8 fields. Whiskers, max and min over the fields.

that many of the most up-regulated genes following DUX4 expression in normal myoblasts are involved in the ubiquitin–proteasome system, including numerous E3 ubiquitin ligases (*Geng et al., 2012*). It is therefore tempting to speculate that DUX4-induced dysregulation of the ubiquitin–proteasome system is responsible for triggering UPF1 protein degradation. However, the precise mechanism by which DUX4 induces UPF1 proteolysis, and whether that mechanism is specific to the FSHD disease state, remains to be elucidated.

DUX4-mediated inhibition of NMD may contribute to FSHD pathophysiology through both cell autonomous and non-cell autonomous mechanisms. The accumulation of abnormal RNAs may cause direct or indirect toxic effects in muscle cells due to intrinsic toxicity of abnormal RNAs or a stress response to the production of abnormal proteins. FSHD muscle is frequently characterized by a T-cell infiltrate (*Arahata et al., 1995*), and it is possible that stabilized NMD substrates encode novel peptides with antigenic potential, contributing to an immune response (*Pastor et al., 2010*). Production of

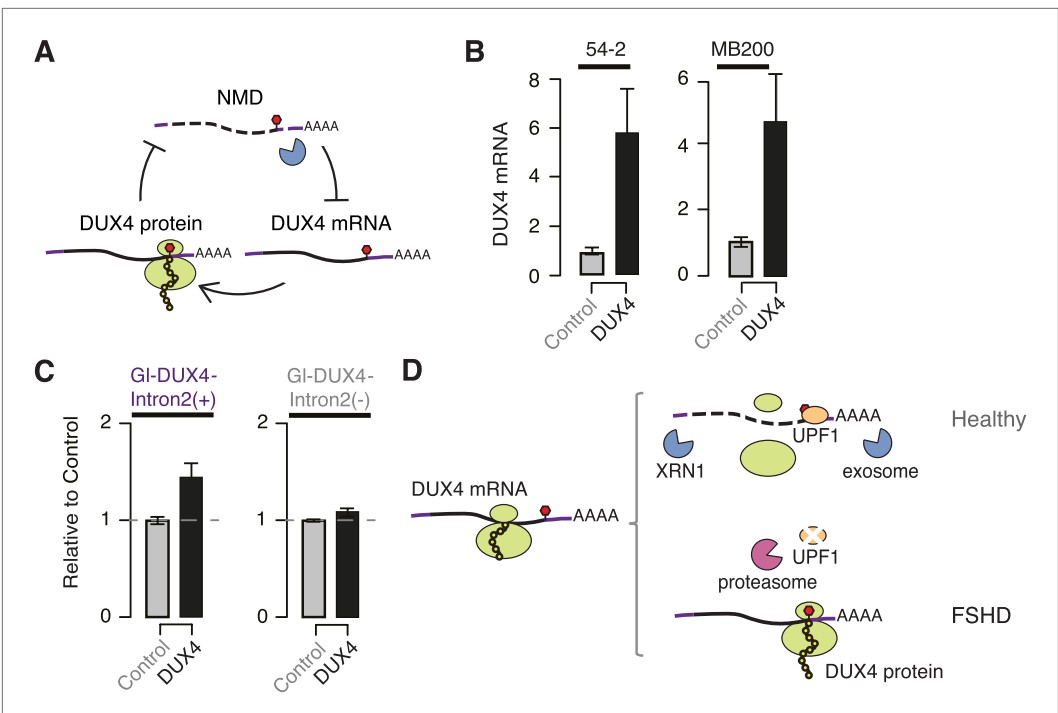

**Figure 4**. *DUX4* and NMD form a feedback loop. (**A**) Schematic of potential double-negative feedback loop between *DUX4* and NMD, in which DUX4 inhibits NMD and NMD degrades DUX4 mRNA. (**B**) Levels of endogenously transcribed DUX4 mRNA following control treatment or ectopic *DUX4* expression, measured 2 days after the initiation of myogenesis in FSHD1 (54-2) and FSHD2 (MB200) cells. (**C**) Relative levels of transcripts from the Gl-DUX4-Intron2(+) and Gl-DUX4-Intron2(−) constructs following control treatment or ectopic *DUX4* expression in normal myoblasts (54-1 cells). For each construct, data are normalized such that the siControl treatment is 1. Error bars, standard deviation. (**D**) Schematic of potential model of interactions between DUX4 and NMD in healthy (top) and FSHD (bottom) muscle cells. In healthy cells, DUX4 mRNA is efficiently degraded by NMD; in FSHD cells, DUX4 triggers proteolytic degradation of UPF1 and inhibits NMD, resulting in the accumulation of DUX4 mRNA and protein.

antigenic peptides could potentially enable even a small fraction of DUX4+ nuclei to induce widespread pathology within a muscle fiber. Directly detecting or measuring DUX4-induced NMD inhibition in FSHD muscle biopsies or in bulk populations of cultured FSHD muscle cells is not feasible due to the low fraction of DUX4+ nuclei present at any given time in the absence of ectopic *DUX4* expression. With DUX4+ nuclei constituting only 0.3% of the bulk population of cultured FSHD muscle cells (*Figure 3G*), changes in the ratios of NMD and non-NMD isoforms in these DUX4+ nuclei are swamped by the normal levels expressed by the vast majority of DUX4− nuclei. Single-cell assays of NMD efficiency are likely required to demonstrate DUX4-induced NMD inhibition in unperturbed patient cells. However, future efforts to identify the downstream antigenic products or toxic effects of stabilized NMD substrates may prove fruitful even in a bulk cell population.

Consistent with the idea that NMD inhibition may contribute to DUX4 toxicity in skeletal muscle, it is interesting to note that the degree of NMD inhibition induced by DUX4 is comparable to that observed in previous studies involving genetic ablation of components of the NMD machinery. For example, a recent study of mouse embryonic fibroblasts lacking *Smg1*, which encodes a kinase responsible for phosphorylating UPF1, found that 9% of predicted NMD substrates created by alternative splicing exhibited increased levels relative to wild-type cells (*McIlwain et al., 2010*). In comparison, we found that 13% of such substrates were up-regulated following DUX4 expression (*Figure 1*), suggesting that DUX4-induced NMD inhibition causes profoundly abnormal RNA metabolism. As RNA toxicity is the major pathophysiologic mechanism in myotonic dystrophy, it is interesting to consider that RNA-mediated disease mechanisms may also have important roles in FSHD.

## Materials and methods

### Accession codes

FASTQ files for the *DUX4* expression experiments were downloaded from the NCBI GEO database under accession number GSE45883 (*Young et al., 2013*).

### Genome annotations

The UCSC knownGene (*Meyer et al., 2013*) and Ensembl 71 (*Flicek et al., 2013*) genome annotations were merged to create a single genome annotation. Splicing event annotations from MISO v2.0 (*Katz et al., 2010*) were then added to this merged genome annotation. Constitutive splice junctions were defined as those for which neither the 5′ nor 3′ splice site was alternatively spliced in the UCSC known-Gene annotation. Predicted NMD substrates were annotated by identifying isoforms containing premature termination codons >50 nt upstream of a splice junction. For the purposes of predicting NMD substrates, open reading frames were assigned based on UniProt annotations (*UniProt Consortium, 2012*) when available, and Ensembl predicted reading frames when UniProt annotations were not available. For the purposes of RNA-seq read mapping, an additional annotation file consisting of all splice junctions annotated in the UCSC, Ensembl 71, and MISO v2.0 annotations was created. This splice junction file was then with a list of all possible junctions between the annotated 5′ and 3′ splice sites of isoforms in the annotation (to detect novel alternative splicing).

### RNA-seq read mapping

Reads were mapped to the UCSC hg19 (NCBI GRCh37) genome assembly. RSEM (*Li and Dewey, 2011*) was modified to call Bowtie (*Langmead et al., 2009*) v1.0.0 with the -v 2 argument. RSEM was then called with the arguments --bowtie-m 100 --bowtie-chunkmbs 500 --calc-ci --output-genome-bam on the genome annotation. Read alignments with mapq scores of 0 and/or a splice junction overhang of less than 6 bp were then filtered out. Remaining unaligned reads were then aligned TopHat (*Trapnell et al., 2009*) v2.0.8b with the arguments --bowtie1 --read-mismatches 2 --read-edit-dist 2 --no-mixed --no-discordant --min-anchor-length 6 --splice-mismatches 0 --min-intron-length 10 --max-intron-length 1000000 --min-isoform-fraction 0.0 --no-novel-juncs --no-novel-indels --raw-juncs on the splice junction file (--mate-inner-dist and --mate-std-dev were calculated by mapping to constitutive coding exons with MISO's exon_utils.py utility). Alignments produced by this call to TopHat were then filtered identically to the alignments produced by RSEM. Reads aligned by RSEM and TopHat were then merged to create BAM files of all aligned reads.

### Gene expression and isoform ratio measurements

Gene expression was quantified using RSEM as described above. Isoform ratios were quantified using two distinct methods. First, MISO was used to quantify isoform ratios for alternative splicing events contained in MISO's v2.0 annotations. Second, novel alternative splicing or intron retention of annotated constitutive splice junctions was quantified using reads crossing the 5′ or 3′ splice sites as previously described (*Hubert et al., 2013*). Differentially spliced events were defined as those with at >20 identifying reads (identifying reads support one or more, but not all, isoforms of a splicing event), a change in isoform ratio ≥10%, and a Bayes factor ≥5 (computed with Wagenmakers's framework [*Wagenmakers et al., 2010*]).

### Viral infection and cell culture

We used previously described lentiviral constructs expressing full-length *DUX4* or GFP as a control (*Geng et al., 2012*). Lentiviral particles were generated by the FHCRC Core Center of Excellence in Hematology Vector Production Core. Viral particle number was estimated with the WPRE element within the viral vector. Myoblasts were transduced at a MOI of ~15 in the presence of 8 μg/ml polybrene. At this MOI, >85% of myoblasts were DUX4+ or GFP+. Unless otherwise noted, cells were collected for analysis 24 hr post-infection. Proliferating myoblasts were cultured in F-10-based growth media (Gibco, Carlsbad, CA) with 20% fetal bovine serum (Gibco) and 1% penicillin/streptomycin (Gibco), supplemented with 10 ng/ml rhFGF (Promega, Madison, WI) and 1 μM dexamethasone (Sigma, St. Louis, MO). Growth media was changed every other day, and proliferating myoblasts were cultured at ≤50% confluence. To initiate myogenic differentiation, myoblasts were switched to an F-10-based differentiation media containing 1% horse serum (Gibco) and 1% penicillin/streptomycin, supplemented with 10 μg/ml insulin (Sigma) and 10 μg/ml transferrin (Sigma) at 99% confluence.

## Plasmid and siRNA transfection

The β-globin NMD(−) and NMD(+) plasmids were previously published as pmCMV-Gl Norm and pmCMV-Gl 39Ter (*Zhang et al., 1998*). Plasmid reporters were transfected with Lipofectamine 2000 (Life Technologies, Carlsbad, CA), unless otherwise noted. To control for transfection efficiency, a control plasmid phCMV-MUP was co-transfected with the reporter as previously described (*Zhang et al., 1998*). 2 µg of reporter along with 500 ng of control plasmid was used for transfecting cells in a six-well format. To measure DUX4-induced changes in NMD efficiency, cells were infected with lentiviral *DUX4* or GFP 24 hr after transfection of the NMD reporters. For the *DUX4* time course experiments, the NMD(−) and NMD(+) reporters were transfected along with phCMV-MUP using the SuperFect reagent (Qiagen, Valencia, CA), and the lentiviral transduction was performed 12 hr post-transfection. siRNAs against UPF1 (Thermo Scientific, Waltham, MA, On-Target siRNA #J-011763-07) and TP53 (Ambion, Silencer Select siRNA #4390824), as well as the control siRNA (Thermo Scientific), were transfected with Lipofectamine RNAiMAX (Life Technologies).

## RNA isolation, real-time qPCR, and endogenous DUX4 mRNA measurement

Cells were lysed with TRIzol (Invitrogen, Carlsbad, CA) and the RNA was extracted according to the manufacturer's instructions. RNA was subsequently cleaned up with Qiagen RNeasy columns, with on-column DNase digestion. 1 µg of RNA was used for cDNA synthesis with Life Technologies SuperScriptIII First-Strand System. 2% of this cDNA was used as a template for real-time qPCR with Life Technologies Power SYBR Green Master Mix. qPCR primer sequences are provided in *Supplementary file 1*. Note that levels of endogenous DUX4 mRNA following ectopic DUX4 expression were measured with primers specific to the DUX4 mRNA's 3′ UTR (the DUX4 lentiviral construct contained the coding sequence alone). To determine how *UPF1* KD affected *DUX4* expression, proliferating 54-2 or MB200 myoblasts were transfected with siUPF1 or siControl and switched to differentiation media 48 hr post-transfection.

## Western blotting

Protein extracts from the *UPF1* KD experiments were generated by lysing cell pellets in protease inhibitor cOmplete ULTRA (Roche, Switzerland) containing RIPA buffer (Cell Signaling Technology, Danvers, MA) along with sonication. For the *DUX4* time course and MG132 treatment, protein was extracted in parallel with the RNA from cells lysed in the TRIzol reagent. Protein pellets were resuspended in a sample buffer containing 5% SDS and 0.5 M unbuffered Tris base to ensure efficient solubilization. Protein concentrations were determined using the Bradford or BCA protein assay. 5 µg of total protein was used for Western blotting. Antibodies used in this study are: anti-UPF1 (Bethyl Laboratories, Montgomery, TX), anti-α-tubulin (Sigma), anti-H3 (Abcam, England), anti-UPF3B (Bethyl Laboratories), anti-SMG7 (Santa Cruz, Dallas, TX). HRP-conjugated (Jackson ImmunoResearch, West Grove, PA) secondary antibodies were used for protein detection in all experiments except for the time course and proteasome inhibitor studies (*Figure 2*). For the experiments reported in *Figure 2*, immunoblotting was performed using the LICOR system with the Odyssey blocking buffer and IRDye-conjugated secondary antibodies (LICOR, Lincoln, NE). Quantification was performed using ImageQuant software (GE Healthcare, Cleveland, OH) using the nonspecific band as a normalizer to account for differences in protein loading. Histone 3 served as an additional loading control, though its very high signal intensity made it an inappropriate normalizer for quantitative analyses.

## Proteasome inhibition

54-1 cells transduced with *DUX4* or GFP lentivirus were treated 16 hr post-infection with 10 µM proteasome inhibitor MG132 (Sigma). Samples were collected 8 hr after MG132 treatment, and UPF1 levels were estimated by immunoblotting. Histone H3, which has a long half-life (*Toyama et al., 2013*), was used as a loading control, in addition to the nonspecific band.

## Fluorescence microscopy and quantification

Cells were permeabilized with PBS containing 0.5% Triton X-100, rinsed in PBS, and blocked in 1% BSA. Primary antibody against DUX4 (Abcam, ab124699) was diluted in blocking buffer at 1:500, and secondary anti-Rabbit TRITC (Jackson ImmunoResearch, 711-025-152) was diluted in blocking buffer at 1:400. For assaying apoptosis in *DUX4*-cells, Image-iT LIVE Red Poly Caspases Detection Kit (Life Technologies, I35101) was used. For both experiments, fluorescently labeled cells were then

viewed under the ZEISS Axiophot fluorescence microscope. For each sample, pictures from eight random fields were taken. ImageJ (Fiji) was used for image analysis and quantification.

## Cloning of chimeric β-globin + DUX4 3′ UTR constructs

The genomic locus of the *DUX4* 3′ UTR (containing both introns) was amplified from a genomic fragment harboring 2.5 D4Z4 repeats (*Gabriëls et al., 1999*) (L42 clone; GenBank ID FJ439133.1). The *DUX4* 3′ UTR lacking the second intron was amplified from cDNA isolated from differentiated 54-2 cells (the first intron is frequently retained in DUX4 cDNA). The β-globin NMD(−) reporter backbone was linearized by forward and reverse PCR primers sitting downstream and upstream of the β-globin 3′ UTR (primers listed in *Supplementary file 1*). Amplicons of the *DUX4* 3′ UTR containing or lacking the second intron were flanked with sequences overlapping the linearized β-globin NMD(−) backbone lacking the β-globin 3′ UTR. The NEB Gibson Assembly Cloning Kit was used to insert the *DUX4* 3′ UTR fragments into the linearized β-globin NMD(−) backbone (New England Biolabs, Ipswich, MA).

## Acknowledgements

We thank Lynne Maquat for sharing the β-globin NMD reporter and MUP transfection control plasmids. This research was supported by the Damon Runyon Cancer Research Foundation DFS 04-12 (RKB), the Ellison Medical Foundation AG-NS-1030-13 (RKB), Fred Hutchinson Cancer Research Center institutional funds (RKB), Friends of FSH Research (SJT), NIH/NIAMS R01AR045203 (SJT), and NIH/NINDS P01NS069539 (SJT).

## Additional information

### Funding

| Funder | Grant reference number | Author |
| --- | --- | --- |
| Damon Runyon Cancer Research Foundation | DFS 04-12 | Robert K Bradley |
| Ellison Medical Foundation | AG-NS-1030-13 | Robert K Bradley |
| Friends of FSH Research | | Stephen J Tapscott |
| National Institutes of Health | R01AR045203 | Stephen J Tapscott |
| National Institutes of Health | P01NS069539 | Stephen J Tapscott |

The funders had no role in study design, data collection and interpretation, or the decision to submit the work for publication.

### Author contributions

QF, Performed NMD inhibition experiments and computational analyses, and created *Figures 1, 3, and 4*; LS, Performed NMD inhibition experiments; SJ, Identified UPF1 proteolysis and created *Figure 2*; RT, SMM, Provided critical biological reagents and information; SJT, Provided overall oversight of the project; RKB, Performed computational analyses and provided overall oversight of the project

### Author ORCIDs

Robert K Bradley, ⓘ http://orcid.org/0000-0002-8046-1063

## Additional files

### Supplementary file

• Supplementary file 1. Primer sequences used for cloning and real-time PCR assays. List of primers used for cloning and real-time PCR. GENE ID, Ensembl ID for the tested genes. RPL27 and SRP14: housekeeping genes. HNRNPD, SRSF2, and SRSF3: inclusion primer sets measure levels of endogenously produced NMD-degraded isoforms. Endogenous DUX4 mRNA was measured with primers specific to its 3′ UTR (RKB_929 and RKB_930). Expression from chimeric Gl-DUX4-Intron2(+) and Gl-DUX4-Intron2(−) constructs was measured with primers specific to the spliced β-globin region.

## Major dataset

The following previously published dataset was used:

| Author(s) | Year | Dataset title | Dataset ID and/or URL | Database, license, and accessibility information |
|---|---|---|---|---|
| Young JM, Whiddon JL, Yao Z, Kasinathan B, Snider L, Geng LN, Tawil R, van der Maarel SM, Tapscott SJ | 2013 | Retroelements and DUX4 Create Primate-specific Promoters for Germline Genes | http://www.ncbi.nlm.nih.gov/geo/query/acc.cgi?acc=GSE45883 | Publicly available at NCBI Gene Expression Omnibus. |

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
