## [Decision Letter]

Thank you for sending your work entitled “A feedback loop between
*DUX4* and nonsense-mediated decay” for consideration at
*eLife*. Your article has been favorably evaluated by James Manley
(Senior editor), a Reviewing editor, and 2 reviewers.

The Reviewing editor and the reviewers discussed their comments before we reached this
decision, and the Reviewing editor has assembled the following comments to help you
prepare a revised submission.

In this short paper, the authors provide evidence for reduced NMD activity playing a
role in Facioscapulohumeral muscular dystrophy (FSHD). Previous work had shown that
mis-expression in skeletal muscle of the double homeobox transcription factor
*DUX4* causes FSHD. However, little was known about the pathways by
which *DUX4* expression in muscle cells leads to the muscle damage
observed in FSHD. Here, the authors show that ectopic expression of
*DUX4* in immortalized and primary myoblast cell lines results in
increased levels of transcripts that are known or predicted to be targets of NMD,
suggesting an overall reduction of NMD activity. The authors speculate that this reduced
NMD activity might lead to the expression of aberrant peptides in muscles that could
either be toxic or activate an immune response against the muscle cells. The authors
further show that concomitant with *DUX4* expression the abundance of
UPF1 protein is reduced (but not the mRNA). Since treatment of the cells with the
proteasome inhibitor MG132 prevented the *DUX*4-induced UPF1 protein
reduction, while MG132 had no effect on UPF1 abundance in cells that did not express
*DUX4*, the authors concluded that *DUX4* expression in
muscle cells causes the proteasome-mediated degradation of UPF1.

In the second part of the paper the authors point out that the two *DUX4*
mRNA isoforms both contain introns in the 3' UTR, which is a well-known feature of
NMD targets. They go on the show that the *DUX4* mRNA is indeed targeted
by NMD and that its 3' UTR is sufficient to induce NMD of a reporter gene. Together
with the first part of the data, this results provide a possible explanation how
occasional bursts of *DUX4* transcription can trigger and enhance
*DUX4* mis-expression in these myoblast regions by an autoregulatory
feedback loop that gradually inactivates NMD in the surrounding nuclei:
*DUX4*-induced UPF1 degradation lowers NMD activity, which in turn
leads to more *DUX4* mRNA and hence protein and thus further
downregulation of NMD. This is an attractive model, and the data for the feedback loop
between *DUX4* expression and NMD is compelling.

Overall, the manuscript is of considerable and broad interest and worthy of publication
in *eLife*. However, there were several substantial concerns of the
reviewers that should be addressed at least in the language of the manuscript:

1) The effects were overall modest, in particular in experiments with the 3' UTR
intron and in the assessment of the fraction of nuclei expressing *DUX4*
protein in FHSD myotubes.

2) Lack of connection between observations in primary myoblast cell lines with
overexpressed *DUX4* and with authentic patient samples—perhaps
too difficult to address, but perhaps it would be possible to ask whether authentic FSHD
muscle cells contain increased proportions of predicted NMD substrates.

3) Lack of a potential molecular mechanism to explain how overexpression of a
transcription factor leads to selective degradation of a set of factors involved in NMD.
One possibility would be to more thoroughly evaluate the levels of other NMD factor
(UPF2 etc.) or components of the proteasome itself. Without elucidation of some
mechanistic link, this part of the story remains purely correlative and therefore less
impactful.

Other more specific points follow:

1) Why is only a small fraction (13%) of predicted NMD targets affected by this
mechanism?

2) Figure 2: Can the authors explain the
transient nature of the upregulation of the PTC-containing beta-globin NMD reporter? In
the light of the proposed feedback loop, one would anticipate robust and enduring
inhibition of NMD (as is observed for *SRSF2* and *SRSF3*
transcripts; Figure 2) and thus the beta-globin
PTC+ levels to remain elevated after 20 hours post transduction with the
*DUX4* expressing lentivirus.

3) Figure 2: The data points for the half-life
measurement of *DUX4* mRNA under siControl conditions scatter quite a
lot; the authors should describe how they did the curve fitting. The conventional way
would be to plot the RNA levels logarithmically and calculate the linear regression line
through these data points. Furthermore, by limiting the time course to only 2 hours, the
extrapolated half-life of 7.86 h in UPF1-depleted cells is most likely highly
inaccurate. The time course should be expanded to 6 or 8 hours to get a more precise
estimate of this half-life.

---

## [Author Response]

*1) The effects were overall modest, in particular in experiments with the
3' UTR intron and in the assessment of the fraction of nuclei expressing DUX4
protein in FHSD myotubes*.

We agree that the experiments with the chimeric Gl-DUX4-Intron2 constructs generally
yielded modest effect sizes, such as increases of 2- and 1.4-fold following
*UPF1* KD or ectopic *DUX4* expression (Figures 3 and 4). These effects were
substantially smaller than the corresponding 4- and 5-fold increases in endogenous
*DUX4* mRNA levels that we observed following *UPF1* KD
or ectopic *DUX4* expression. We are unsure why the chimeric construct
exhibited more modest effect sizes than did the endogenous *DUX4* mRNA,
although it may be due to a combination of its plasmid-based nature and the absence of
sequence features of the *DUX4* mRNA other than its 3' UTR intron
that contribute to its degradation by NMD. As the Gl-DUX4-Intron2 constructs are
chimeric, we feel that the most realistic estimates of effect sizes stem from our
measurements of the endogenous *DUX4* mRNA rather than the
Gl-DUX4-Intron2 constructs.

Outside of the chimeric construct, however, many of the other changes that we observed
were quite dramatic. For example, the reviewers mention the fraction of
*DUX4*+ nuclei in a FSHD myotube culture following control or
UPF1 knock down. *DUX4* exhibits a highly variegated expression pattern,
with *DUX4* detectable in only 0.1-1% of FSHD muscle cells (Snider et
al., PLoS Genetics, 2010; Figure 3 from this
manuscript). Following UPF1 knock down, we found that the fraction of
*DUX4*+ nuclei increased from 0.3% to 2.1%, a change of an order
of magnitude. As the extremely low fraction of *DUX4*+ cells in FSHD
muscle is a defining characteristic of the disease, we feel that an order-of-magnitude
increase in the fraction of *DUX4*+ cells is a substantial effect
size, consistent with NMD playing an important role in the regulation of
*DUX4* levels. We have emphasized this effect size in the revised
manuscript, and also updated Figure 3 to show
additional microscopy fields.

To place these results in context, it is useful to compare our observations with
previous studies where NMD was directly inhibited. For example, a recent study (Hurt et
al., Genome Research, 2013), inhibited NMD with shRNAs against UPF1 or cycloheximide
treatment, which prevents protein translation and thereby NMD. Hurt et al. then used
RNA-seq to quantify global mRNA levels. Hurt et al. identified only 8, 14, or 40
predicted NMD substrates out of >20,000 queried mRNAs that were up-regulated by at
least 4-fold in cells treated with one of two distinct shRNAs against UPF1 (8 and 14
mRNAs identified) or cycloheximide (40 mRNAs identified). In the context of these
results, we feel that the 4- and 5-fold increases in endogenous *DUX4*
mRNA levels that we observe following UPF1 knock down or ectopic *DUX4*
expression are quite substantial.

As discussed further below in specific point #1, we also note that the degree of
NMD inhibition that is caused by *DUX4* expression is quite remarkable,
matching the degree of NMD inhibition caused by genetic knock out of essential NMD
factors.

*2) Lack of connection between observations in primary myoblast cell lines with
overexpressed DUX4 and with authentic patient samples—perhaps too difficult to
address, but perhaps it would be possible to ask whether authentic FSHD muscle cells
contain increased proportions of predicted NMD substrates*.

We agree that our conclusions would be even further strengthened by a direct
demonstration that *DUX4* causes NMD inhibition in primary patient
materials, in the absence of ectopic *DUX4* expression. However,
*DUX4*’s variegated expression pattern—with
*DUX4* detectable in only 0.1-1% of FSHD muscle cells (Snider et al.,
PLoS Genetics, 2010; Figure 3 from this
manuscript)—typically renders such experiments infeasible. For this reason, most
studies of *DUX4*’s molecular function have relied upon ectopic
expression to ensure that the majority of assayed cells are
*DUX4*+.

The challenge of detecting *DUX4*-induced NMD inhibition given
*DUX4*’s variegated expression pattern can be readily seen with
a statistical power estimation. For example, we used a poison exon-containing isoform of
*SRSF3* as an endogenous marker of NMD. This isoform increased
dramatically in abundance (from ∼10% to 65% of *SRSF3*’s
mRNA population) following ectopic *DUX4* expression. It is therefore
reasonable to expect that *DUX4*+ cells in a FSHD muscle cell
culture may similarly express abnormally high levels of this *SRSF3* NMD
substrate in the absence of ectopic *DUX4* expression. However, the
variegated nature of *DUX4* expression prevents the direct detection of
such signals of NMD inhibition simply due to statistical power limitations.
*DUX4* expression is detectable in 0.3% of FSHD nuclei by
immunostaining (Figure 3). Assume that the
*SRSF3* NMD substrate is present at isoform ratios of 10% and 65% in
*DUX4*- and *DUX4*+ cells, respectively.
Measurements of the *SRSF3* NMD substrate in a bulk FSHD cell population
would therefore yield an isoform ratio of (0.003 x 65%) + (0.997 x 10%) =
10.165%. The difference between an isoform ratio of 10% for *DUX4*- cells
and 10.165% for a mixture of *DUX4*+ and *DUX4*-
cells is not detectable with RNA-seq or quantitative PCR.

In the future, we may be able to overcome the challenges presented by
*DUX4*’s variegated expression pattern by studying single cells
with assays such as single-cell RNA-seq or live cell imaging approaches. However, we
feel that such assays are beyond the scope of this manuscript. We have added additional
text to the discussion in which we briefly describe the challenges with detecting NMD
inhibition in a bulk population of FSHD cells.

*3) Lack of a potential molecular mechanism to explain how overexpression of a
transcription factor leads to selective degradation of a set of factors involved in
NMD. One possibility would be to more thoroughly evaluate the levels of other NMD
factor (UPF2 etc.) or components of the proteasome itself. Without elucidation of
some mechanistic link, this part of the story remains purely correlative and
therefore less impactful*.

We agree that this aspect of the manuscript is correlative. Given the close temporal
connection between decreased UPF1 protein levels and increased levels of NMD substrates,
we do feel that it is reasonable to speculate that UPF1 degradation is a likely
contributor to NMD inhibition. In the revised manuscript, we are careful to state
clearly that the data is suggestive but purely correlative. Nonetheless, as rapid UPF1
protein degradation has not been previously described to our knowledge, we feel that
this section of the manuscript is a significant addition to the field, as it suggests a
potential new mechanism for the cell to alter RNA surveillance efficiency.

In a previous study (Geng et al., Developmental Cell, 2012), we noticed that many genes
that were differentially expressed following *DUX4* expression were
involved in protein ubiquitination, including numerous E3 ubiquitin ligases. It is
possible, for example, that UPF1 is a substrate of one of the many E3 ligases that are
up-regulated in *DUX4*-expressing muscle cells. While we do hope to
eventually determine how *DUX4* expression triggers UPF1 degradation, we
feel that such assays are beyond the scope of this manuscript given the many different
ways in which a transcription factor could indirectly influence UFP1 proteolysis. We
briefly discuss this in the revised manuscript.

*Other more specific points follow*:

1) Why is only a small fraction (13%) of predicted NMD targets affected by this
mechanism?

We agree that it is interesting that only a minority of premature stop codon-containing
mRNAs exhibit increased levels following *DUX4* expression and
concomitant NMD inhibition. This could be due to imperfect annotation of NMD substrates,
differential sensitivity of different NMD substrates to reduced UPF1 levels, or other
factors.

It is important to note that this 13% is quite high when compared to other studies where
NMD has been inhibited genetically or with RNAi. For example, a recent study (McIlwain
et al., PNAS, 2010) used a gene trap to generate mouse embryonic fibroblasts lacking
*Smg1*, which is responsible for UPF1 phosphorylation, a required step
in NMD. McIlwain et al. then used RNA-seq to characterize the transcriptomes of
*Smg1*-deficient cells, finding that “in the absence of
*Smg1*, approximately 9% of predicted PTC-containing AS events
displayed changes of more than 10%, and approximately 2% showed changes of more than
20%” (PTC = premature termination codon; AS = alternative splicing).
Other studies conducted in worms, flies, and mammalian cells found that similarly small
fractions of predicted NMD substrates were affected by depletion of core NMD
factors.

Therefore, we agree that it is interesting that only 13% of predicted NMD substrates are
up-regulated in *DUX4*-expressing cells. However, as this degree of
up-regulation is comparable to that observed in previous studies where NMD was directly
inhibited genetically, we believe that this 13% is a remarkably high fraction. We
comment on this in the revised manuscript.

*2)*
Figure 2*: Can the
authors explain the transient nature of the upregulation of the PTC-containing
beta-globin NMD reporter? In the light of the proposed feedback loop, one would
anticipate robust and enduring inhibition of NMD (as is observed for* SRSF2
*and* SRSF3 *transcripts;*
Figure 2*) and thus the
beta-globin PTC+ levels to remain elevated after 20 hours post transduction with
the DUX4 expressing lentivirus*.

As the reviewers point out, levels of the NMD(+) β-globin reporter increase
at the same time (∼12 h following infection) as the endogenous
*SRSF2* and *SRSF3* markers, but the NMD(+)
β-globin reporter up-regulation does not last for the entire time course. This is
surprising because (1) *SRSF2* and *SRSF3* NMD substrates
exhibit increased levels throughout the time course, (2) UPF1 levels decrease
monotonically throughout the time course, and (3) our RNA-seq data from MB135 and 54-1
cells was obtained at 24 hours and 48 hours time points, respectively, and both samples
exhibited obvious global increases in NMD substrate levels. Therefore, while we are
confident that NMD inhibition occurs throughout the time course, we are unsure why this
enduring behavior is not reflected in the NMD(+) β-globin reporter. We are
unaware of a previous study that has used this reporter to study the dynamics of NMD
efficiency along a time course. For unclear reasons, such a plasmid-based reporter may
be a more useful tool for measuring NMD efficiency in a static, rather than dynamic,
context.

*3)*
Figure 2*: The data
points for the half-life measurement of DUX4 mRNA under siControl conditions scatter
quite a lot; the authors should describe how they did the curve fitting. The
conventional way would be to plot the RNA levels logarithmically and calculate the
linear regression line through these data points*. *Furthermore, by
limiting the time course to only 2 hours, the extrapolated half-life of 7.86 h in
UPF1-depleted cells is most likely highly inaccurate. The time course should be
expanded to 6 or 8 hours to get a more precise estimate of this
half-life.*

To estimate the half-life of *DUX4* mRNA, we computed the best-fit
exponential model using untransformed data, which is mathematically identical to the
reviewers’ suggestion of computing a best-fit linear model for log-transformed
data. We apologize for not explaining this half-life estimation procedure more clearly
in the initial submission.

We agree that the data points exhibit substantial variation, particularly for the
siControl treatment. Unfortunately, this high level of variability is an unavoidable
consequence of *DUX4*’s variegated expression. Since
*DUX4* is detectable in only 0.3% of nuclei by immunostaining for the
siControl treatment (Figure 3),
*DUX4* mRNA levels are very low even in large cultures of FSHD cells.
Estimates of *DUX4* mRNA levels therefore are noisy and become
increasingly so following transcription shutoff.

We agree with the reviewers that extrapolating a ∼8 hour half-life from a 2 hour
time course is prone to substantial measurement error. We accordingly repeated the
*DUX4* mRNA half-life measurement multiple times with a longer 8 hour
time course. However, due to the aforementioned difficulty of accurately measuring
*DUX4* mRNA following transcription shutoff even in a large culture of
FSHD muscle cells, we were unfortunately unable to obtain reliable data from the longer
time courses.

As the *DUX4* mRNA half-life measurement was not an essential component
of the manuscript—since we provide multiple other sources of evidence that
*DUX4* mRNA is a NMD substrate, due in part to its spliced 3'
UTR—we have decided to remove the half-life data entirely from the revised
manuscript. We agree with the reviewers that this data is noisy, and feel that it is a
distraction from the otherwise clean data that we present here.